# A simulation based difficult conversations intervention for neonatal intensive care unit nurse practitioners: A randomized controlled trial

Roberta Bowen[1], Kate M. Lally[2,3], Francine R. Pingitore[3,4], Richard Tucker[1], Elisabeth C. McGowan[1,3], Beatrice E. Lechner[1,3]*

1 Department of Neonatology, Women & Infants Hospital, Providence, RI, United States of America,
2 Program in Palliative Care, Care New England Health System, Providence, RI, United States of America,
3 Warren Alpert Medical School of Brown University, Providence, RI, United States of America,
4 Department of Pediatrics, Hasbro Children's Hospital, Providence, RI, United States of America

* blechner@wihri.org

## Abstract

**Data Availability Statement:** All relevant data are within the manuscript.

### Background

Neonatal nurse practitioners are often the front line providers in discussing unexpected news with parents. This study seeks to evaluate whether a simulation based Difficult Conversations Workshop for neonatal nurse practitioners leads to improved skills in conducting difficult conversations.

### Methods

We performed a randomized controlled study of a simulation based Difficult Conversations Workshop for neonatal nurse practitioners (n = 13) in a regional level IV neonatal intensive care unit to test the hypothesis that this intervention would improve communication skills. A simulated test conversation was performed after the workshop by the intervention group and before the workshop by the control group. Two independent blinded content experts scored each conversation using a quantitative communication skills performance checklist and by assigning an empathy score. Standard statistical analysis was performed.

### Results

Randomization occurred as follows: n = 5 to the intervention group, n = 7 to the control group. All participants were analyzed in each group. Participation in the simulation based Difficult Conversations Workshop increases participants' empathy score (p = 0.015) and the use of communication skills (p = 0.013) in a simulated clinical encounter.

**Funding:** The authors received no specific funding for this work.

**Competing interests:** The authors have declared that no competing interests exist.

## Conclusions

Our study demonstrates that a lecture and simulation based Difficult Conversations Workshop for neonatal nurse practitioners improves objective communication skills and empathy in conducting difficult conversations.

## Introduction

The ability to communicate effectively with patients' families is an essential skill for those caring for infants in the neonatal intensive care unit (NICU). Delivering bad news is a skill set not typically taught in the formal education of advanced practice registered nurses. In the United States, these advanced practice registered nurses, or nurse practitioners (NPs), provide care alongside physicians, often in a role similar to the physician's role and sometimes in lieu of the physician. In the NICU, neonatal NPs diagnose and treat infants, perform procedures and interact with and provide support to parents. Thus, acquisition of skills for leading difficult conversations is essential for nurse practitioners to be successful in their full scope of practice. Conducting research on communication skills training in the clinical setting is challenging and the current status of the field does not allow for the identification of one gold standard [1, 2]. Even fewer studies exist in the context of neonatology. Neonatal NPs feel that their education is lacking in this key component of practice [3], and studies of NICU communication skills did not include NPs in the assessment [4] or only measured NPs' self-reported and thus subjective outcomes [5]. The complicated communication task of delivering bad news to the parents of infants is fraught with discomfort and uncertainty for the practitioner delivering the news [6], especially given that bad news around the birth of an infant is not in line with parental expectations.

Most clinicians rely on skills demonstrated by mentors or those learned by trial and error, despite the fact that taking part in a formal program to enhance communication skills leads to an improvement in communication skills [7, 8], while studies have demonstrated that patients desire good communication [9] and that communication skills can be taught and retained [10]. Parents of infants in the NICU are at very high risk for adverse mental health outcomes [11]. Thus, communication approaches used by the medical team, including NPs, gain utmost importance. When working in level 1 and 2 community hospital nurseries, neonatal nurse practitioners are often the front line providers in discussing unexpected news with parents. Thus, we sought to evaluate the hypothesis that a lecture and simulation based Difficult Conversations Workshop for the neonatal nurse practitioners will increase skill in conducting difficult conversations with patients' families.

## Materials and methods

We performed a randomized controlled prospective study of a simulation based Difficult Conversations Workshop for NICU nurse practitioner staff at a large regional level IV NICU in the Northeast of the United States. The research related to human use has been approved by the Women & Infants Hospital Institutional Review Board. Written informed consent was obtained. In this 80 bed level 3 NICU, a simulation based Difficult Conversations Workshop is part of the training program for the neonatal-perinatal medicine fellows.

### Participants

The clinical NICU nurse practitioner group consists of 31 nurse practitioners, who work in a level IV regional NICU as well as multiple level II community hospital NICUs. All NPs were

invited to participate in the study. The NPs were recruited to participate in the study using email as well as a presentation of the study by one of the study authors at a monthly NP staff meeting. Recruitment and workshop were performed from May 2016 to July 2016. Each three hour session of the simulation based Difficult Conversations Workshop consisted of 4–6 participants. Participants in each session were randomized using the web-based randomization tool Randomizer.org to either the intervention or control group. Simple randomization was performed with a randomization allocation of 1:1. Randomization was performed at the beginning of the workshop. Both groups participated in a three hour workshop.

## Study structure (Fig 1)

Prior to randomization, all study participants (intervention group and control group) filled out an anonymous pre-workshop survey. Then, after randomization, the control group performed the Test Scenario, which was a standardized clinically relevant simulation scenario using trained improvisational actors as parents. They then took part in the simulation based Difficult Conversations Workshop so as to allow them the opportunity to benefit from the learning opportunity. The intervention group, on the other hand, took part in the simulation based Difficult Conversations Workshop prior to performing the Test Scenario. At the end of the Workshop and Test Scenarios, all participants filled out a post-workshop survey. Data collection on the pre- and post-workshop surveys ascertained demographics, past experiences with communication skills training, past experiences leading difficult conversations in the clinical setting, as well as feedback on the workshop. The workshop took place in the Care New England Simulation Center at Women & Infants Hospital.

## Simulation based difficult conversations Workshop

The simulation based Difficult Conversations Workshop was a 4.5 hour workshop that consisted of three components (Fig 1). First, the participants were presented with a lecture on difficult conversation communication skills. This lecture was 30 minutes long and highlighted the basic tenets of communication skills in healthcare. Next, each participant took part in a simulation Teaching Scenario, a clinically relevant practice difficult conversation with a trained improvisational actor that was about ten minutes long, while remaining participants observed the scenario via live video. Finally, at the end of the Teaching Scenarios, a facilitated debriefing session was held for all participants. This debriefing session was usually an hour to two hours in length. The workshop was led by a neonatologist who is the director of and trainer in the Difficult Conversations for Neonatal Fellows Training program. Each simulated Teaching Scenario reflected a situation typical of the NICU NP's work environment. The trained actors functioned in the role of a parent during the simulated difficult conversations.

## Performance assessment

The Test Scenario was a 10 minute conversation with a trained improvisational actor in a simulated standardized clinical scenario. The encounter took place in the Women & Infants Hospital Simulation Center and was videotaped, but not shown via live video to any intervention NPs, control NPs or trainers (in contrast to the Teaching Scenarios). This was done to maintain the integrity of the standardized Test Scenario for all intervention and control NPs. The Test Scenario simulation was scored at a later date independently by two blinded content expert observers. One observer was a board certified palliative care physician; the other observer was a doctorally prepared pediatric psychiatric clinical nurse specialist with expertise in interpersonal communication and relationships. These observers did not work with or know any of the participants and were blinded to participant group. In order to assess the

## Study flow diagram

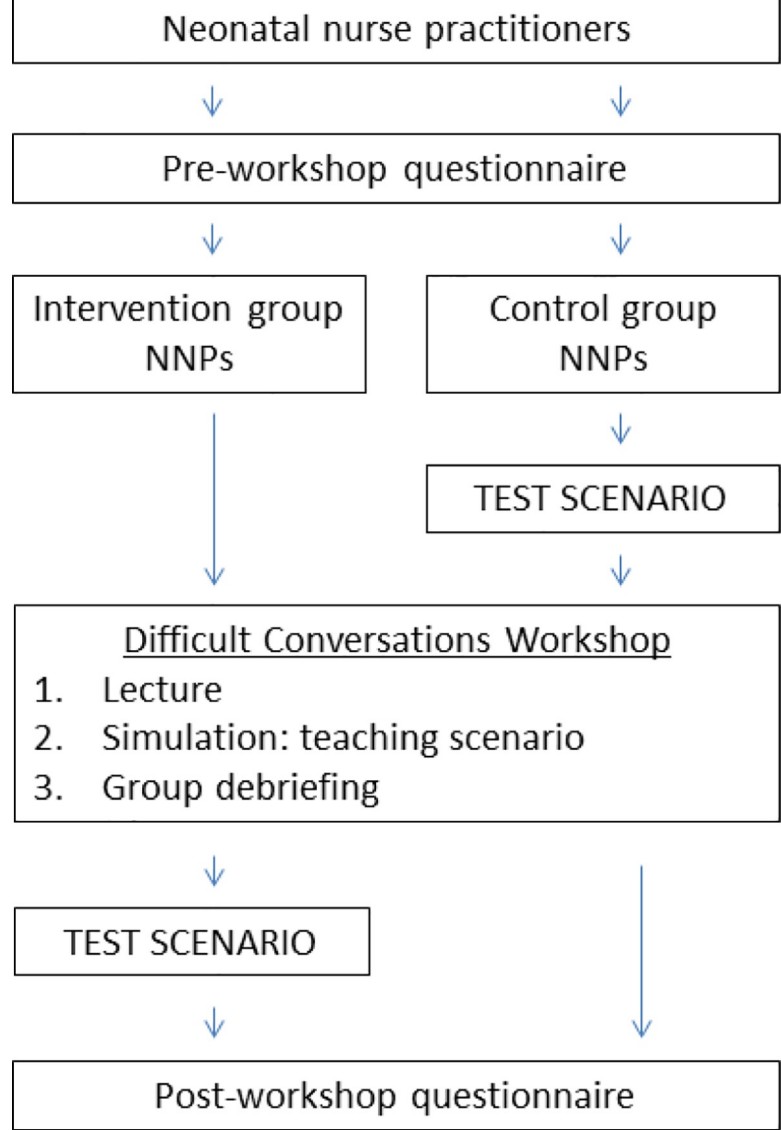

**Fig 1. Study flow diagram.** NNP = neonatal nurse practitioner.

performance of each participant, the observers completed a quantitative communication skills performance checklist as well as assigning an empathy score to rate the participant's level of empathy on a scale of 1 (no empathy) to 10 (extremely empathetic) (Fig 2). The quantitative communication skills performance checklist was developed using a two-step approach. A review of the literature was performed for communication skill checklists, then the final checklist was curated by the authors via expert consensus. The empathy score was developed via expert consensus.

**NNP Difficult Conversation Evaluation Tool**

**Check all skills satisfactorily completed.**

_____    Introduction-introduces/re-introduces self

_____    Comfort-as circumstance allows, assures parents' comfort and ensures privacy

_____    As circumstance allows, asks parent(s) who they would like to be present for the conversation (do visitors need to leave; are both mother and father present)

_____    Body Position (appropriate eye contact, open posture, leans forward toward parents; does not hover over parents)

_____    Uses clear understandable language; no medical jargon ("atypical features" instead of dysmorphic features/Down Syndrome/Trisomy 21)

_____    Uses the baby's name during the conversation

_____    Asks what the parent(s) know/suspect

_____    Foreshadows the bad news ("I'm sorry but I have bad news")

_____    Pauses consciously and allows for silence after delivering bad news

_____    Uses expressions that communicate empathy ("I wish I had better news")

_____    Acknowledges the parents' emotions ("I can see how worried you are," "I know this must be shocking," "It's OK to cry," "I can see that you don't know what to say")

_____    Asks parents open-ended questions

_____    Makes statements that furnish hope

_____    Offers supportive resources for the parents (chaplain, social worker, etc.)

_____    Summarizes and makes a follow up plan. Assures parents they will keep them informed.

_____    Invites parents' questions and questions are answered in a straightforward manner

Empathy Score:

Please rate the NNP's level of empathy on a scale of 1 (no empathy) to 10 (extremely empathetic)

Please circle:   1     2     3     4     5     6     7     8     9     10

**Fig 2. Evaluation tool utilized by blinded independent content experts to evaluate recorded Difficult Conversations Test Scenarios performed by participants.** NNP = neonatal nurse practitioner.

## Teaching & Test Scenarios

In Teaching Scenario #1, a mother was informed that child protective services had informed the hospital that they would investigate the mother after her twins were born. In Teaching Scenario #2, a mother was informed that her infant had failed a congenital heart disease screening and needed to be transferred to a regional NICU to rule out congenital heart disease. In the Test Scenario, a mother was told that there was clinical suspicion of Down Syndrome in her newborn.

## Data analysis

Statistical analysis was performed as follows. Differences in exhibition of communication skills between the groups was tested using Fisher's exact test, and numbers of skills demonstrated and empathy scores were compared via the Student's t-test.

Inter-rater reliability on the scoring of the Test Scenario was measured for the communication skills items using a pooled kappa statistic. Rater agreement on empathy scores was calculated using the two one-sided t-tests (TOST) method, with agreement limits of ±3 points.

## Results

13 out of 31 participated; n = 5 in the intervention group, n = 7 in the control group. One video could not be assessed due to technical difficulties with sound recording. Demographics of the group and experience with difficult conversations as a trainee and in the clinical setting are presented in Table 1.

**Table 1. Participant demographics and experience with difficult conversations.**

| Survey questions | n = 13 (%) |
|---|---|
| Number of years as an NP taking care of infants | |
| 0–1 | 1 (8) |
| 2–5 | 5 (38) |
| 6–10 | 2 (15) |
| > 10 | 4 (31) |
| Average number of weekly hours worked | |
| 12–24 | 0 (0) |
| 25–32 | 2 (17) |
| 33–40 | 3 (25) |
| 41–55 | 3 (25) |
| > 55 | 4 (33) |
| NICU level most often worked in | |
| 3 or 4 | 12 (92) |
| 2 | 1 (8) |
| 1 | 0 (0) |
| Any work in level 1/2 community hospital nursery | |
| yes | 9 (69) |
| Take transport call | |
| yes | 7 (54) |
| Received education during training/career on communicating bad news to the family of an infant | |
| yes | 2 (15) |
| Number of times present in the past year when bad news was given to the family of an infant | |
| 0 | 0 (0) |

(*Continued*)

**Table 1.** (Continued)

| Survey questions | n = 13 (%) |
|---|---|
| 1–2 | 3 (23) |
| 3 or more | 10 (77) |
| Number of times in the past year you gave bad news to the family of an infant | |
| 0 | 3 (23) |
| 1–2 | 5 (38) |
| 3 or more | 5 (38) |
| Extent to which you feel competent to deliver bad news to a family of an infant in your care | |
| not competent | 2 (15) |
| somewhat competent | 7 (54) |
| moderately competent | 3 (23) |
| competent | 1 (8) |
| don't know | 0 (0) |

*Participation in the simulation based Difficult Conversations Workshop increases the use of communication skills in a simulated clinical encounter and increases participants' empathy score.*

In the intervention group, the mean number of predefined communication skill behaviors exhibited by each participant was higher than in the control group (12 skills compared to 8 skills per scenario; p = 0.013). Among the individual communication skill behaviors compared individually between the groups, only "asks parents open-ended questions" was significantly higher in the post intervention group (p = 0.047). In the intervention group, the mean empathy score was higher compared to the control group (8.4 compared to 6.2; p = 0.015).

*Independent of participation in the simulation based Difficult Conversations Workshop, some communication skills are used more often than others.*

The frequency with which individual communication skills were applied in simulated clinical encounters was similar among the two groups. Some skills, such as "Introduces/Re-introduces self", were almost always displayed, while others, such as "Asks parents to repeat back" were never displayed (Table 2).

Interobserver agreement between the two independent blinded reviewers in communication skill scores was 74% agreement overall (range for individual participants between 59 and 94% and for individual skills between 42% and 100%) with an interrater reliability pooled kappa of 0.77. In the empathy score, the top three scores and the bottom two were identical between the two reviewers, while there were some differences in the middle of the field. The two one sided t-tests demonstrated equivalence of empathy score differences, with differences within three points considered equivalent (p = 0.0030).

On the post-intervention survey, participants rated the workshop between 5.8 and 6.0 on a variety of measures on a 6.0 scale (Table 3).

## Discussion

Our study demonstrates that an intervention consisting of a structured lecture and simulation based communication skills workshop for neonatal NPs leads to an increase in the use of specific communication skills as well as improvement in a perceived empathy score in a simulated difficult conversation setting. This is the first study assessing these objective outcomes in neonatal nurse practitioners, while a previous study demonstrated improved self-reported confidence in difficult conversations in neonatal fellows and nurse practitioners [5]. As nurse

**Table 2. Utilized communications skills.**

| Communications skill | Intervention group (n = 5) (%) | Control group (n = 7) (%) |
|---|---|---|
| Introduces/Re-introduces self | 100 | 86 |
| Body Position (Seated/Positioned at eye level to parent; not hovering over parent; lean forward toward parent) | 100 | 86 |
| Makes statements that furnish hope ("I hope I am wrong about this") | 100 | 86 |
| Summarizes and makes a follow up plan. Assures parents they will be available | 100 | 79 |
| Avoids medical jargon ("atypical features" instead of dysmorphic features/Down Syndrome/Trisomy 21) | 100 | 64 |
| Uses expressions that communicate empathy ("I wish I had better news") | 80 | 64 |
| Uses the baby's name during the conversation | 80 | 50 |
| Suggests additional supportive resources for the parents (chaplain, social worker, etc) | 90 | 50 |
| Asks what the parent(s) know/suspect | 50 | 43 |
| Speaks slowly in short simple sentences | 80 | 43 |
| Acknowledges the parents' emotions ("I can see how worried you are," "I know this must be shocking," "It's OK to cry," "I can see that you don't know what to say") | 40 | 43 |
| Asks parents open-ended questions | 80 | 36 |
| Asks parent(s) if there is anyone else they would like to be present for the meeting | 70 | 29 |
| Foreshadows the bad news ("I'm sorry but I have bad news") | 70 | 29 |
| Pauses consciously and allows for silence after delivering bad news | 60 | 21 |
| If visitors present, gives family a choice on who should be present for the meeting | 00 | 00 |
| Asks parents to repeat back what they have been told | 00 | 00 |

practitioners are important members of the multidisciplinary teams providing care for neonates in many NICUs across the United States, it is important to train neonatal NPs in difficult conversations and breaking bad news, particularly in the current changing climate in healthcare, where NPs are providing more and more care in academic medical centers as well as community hospitals.

Objectively assessable specific communication skills are important components of difficult conversations in the clinical setting. Our results are supported by other studies, which have shown that simulation is an effective tool for realistic training in difficult conversations in the context of neonatal care, for example in decision-making at the limits of viability [12], as well as in pediatrics [8], and leads to improved actual communication skills [7]. Studies also

**Table 3. Participant workshop evaluations.**

| Mean score (1–6 [extremely ineffective/unsatisfactory—extremely effective/outstanding]) | n = 13 |
|---|---|
| The lecture on communication skills was helpful/informative | 5.9 |
| The simulation was helpful/informative | 5.9 |
| The facilitated debriefing was helpful/informative | 6.0 |
| The environment felt safe to ask questions/share thoughts | 6.0 |
| After attending the workshop, I feel more competent to lead a difficult conversation | 5.8 |
| Overall satisfaction with the session | 6.0 |
| The workshop should be part of neonatal NP orientation/training | 6.0 |

demonstrate that taking part in a formal program aimed at improving communication skills in difficult conversations influences pediatric provider confidence in managing difficult clinical scenarios [13], as well as leading to better humanistic skills and better delivery of bad news [14] and to improved knowledge and comfort levels in communication [15].

Communication skills training has also been used successfully to improve residents' skills in code status discussions [16], for genetic counseling training [17], communication for anesthesia residents[10] and the disclosure of medical errors [18]. Furthermore, simulation has been shown to improve long term retention of skills and self-reported changes in behavior [19] [15] [20] as well as the long term retention of confidence in one's communication skills in breaking bad news [21].

While it is important to assess objective communication skills in difficult conversations, not every aspect of the level of skill that care providers demonstrate can be assessed using a specific skill checklist. In addition to the objective communication skills that are necessary for breaking bad news, empathy plays a significant role in patient-provider communication. Parents prioritize communication[9] and want caring providers, for example when receiving prenatal consults by neonatologists for congenital anomalies [22]. Additionally, studies have shown that physician empathy is associated with increased adherence to therapy and improved clinical outcomes [23–25]. While possible differences in communication style between physicians and nurse practitioners have not been studied, nurses' and physicians' patterns of communication differ in enacted NICU conversations; physicians provide more biomedical information while nurses provide more psychosocial information [4].

Furthermore, empathy is an important component of the patient-practitioner interaction from the practitioner perspective as well. For example, empathy in medical students is associated with a decrease in burnout [26, 27]. Thus, the empathy score was utilized as an additional marker of the interaction between the provider and patient.

Our study demonstrates that lecture plus simulation based training improves empathy perceived by an expert observer in addition to improving objective communication skills. Empathy does not lend itself to one simple definition. One approach to categorizing empathy is into cognitive empathy vs. affective empathy, reviewed in [28], in which cognitive empathy is associated with external traits that can be learned, while affective empathy is not. Thus, for the purposes of this study, we defined empathy as a cognitive and thus behavioral trait, which consequently is a characteristic that lends itself to modification by training. Despite the abundance of alternative definitions for empathy [28], the two independent content expert observers were able to assess empathy in the Test Scenarios with high interrater reliability. They ranked the level of empathy that participants displayed in scenarios similarly: both their highest and lowest ranking participants were identical, irrespective of the actual number of the empathy score on the scale. Such concordance was achieved despite the fact that the content expert observers were not trained to look for specific signs, but received the sole instructions to score scenarios on a scale of 1 (no empathy) to 10 (extremely empathic).

One limitation of this study is that individual participants were not tested using both a pre- and a post-intervention scenario, given that the increased time commitment necessary for that experimental model was not possible due to participants' clinical staffing requirements. The disadvantage of not having the same participants in both the pre- and post-intervention group is a decrease in the signal to noise ratio. However, given that we nonetheless saw significant improvement in both specific skills and overall empathy scores, we hypothesize that the results would have been even stronger if interpersonal differences had been accounted for using the same participants for both arms of the study.

Another limitation of the study is that the communication skill result and the empathy score result may not be independent variables, as it is possible that the observing content

expert evaluators were subconsciously influenced in their assessment of the empathy score based on the number of communication skills demonstrated. If this were the case, this would not detract from the validity of the results. In fact, this mechanism, if it were at play in these assessments, would support the hypothesis that empathy can be learned as a specific skill set, aligning with the cognitive/behavioral definition of empathy, thus suggesting that specific learned communicative behavioral skills may impact the patient's perception of empathy.

Furthermore, we recognize that only 13/31 NPs took part in this workshop. Upon further investigation, the most common reason for non-participation included the complex scheduling of clinical load. To see if these results are generalizable, a larger cohort may be needed. Nonetheless, this small study demonstrated a difference between the intervention and control groups.

Since others have shown that trainees and their program directors are more lenient in their assessment of communication simulation performance compared to patients and communication experts [29], an advantage of this study is that we utilized independent blinded content experts to perform the video assessments for both the skills assessment and the empathy score.

An additional approach that may improve difficult conversation skill and empathy scores may be to incorporate erroneous examples into the lecture component of the workshop, as these have been shown to improve breaking bad news simulation performance in nursing students [30].

In summary, our study demonstrates that a lecture and simulation based Difficult Conversations Simulation workshop improves objective communication skills and empathy in neonatal nurse practitioners in conducting difficult conversations with patients' families as perceived by an expert observer. Future studies will need to address the long term retention of learned communication skills as well as the transfer of communication skills from simulation settings to actual clinical practice.

## Author Contributions

**Conceptualization:** Roberta Bowen, Elisabeth C. McGowan, Beatrice E. Lechner.

**Data curation:** Roberta Bowen, Kate M. Lally, Francine R. Pingitore, Elisabeth C. McGowan, Beatrice E. Lechner.

**Formal analysis:** Richard Tucker, Elisabeth C. McGowan, Beatrice E. Lechner.

**Investigation:** Roberta Bowen, Kate M. Lally, Francine R. Pingitore, Elisabeth C. McGowan, Beatrice E. Lechner.

**Methodology:** Kate M. Lally, Richard Tucker, Elisabeth C. McGowan, Beatrice E. Lechner.

**Project administration:** Roberta Bowen, Elisabeth C. McGowan, Beatrice E. Lechner.

**Resources:** Beatrice E. Lechner.

**Supervision:** Beatrice E. Lechner.

**Writing – original draft:** Roberta Bowen, Beatrice E. Lechner.

**Writing – review & editing:** Roberta Bowen, Kate M. Lally, Francine R. Pingitore, Richard Tucker, Elisabeth C. McGowan, Beatrice E. Lechner.

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
