## [Decision Letter · Decision Letter 0]

20 Jan 2020

PONE-D-19-33859

A simulation based difficult conversations intervention for NICU nurse practitioners: A randomized controlled trial

PLOS ONE

Dear Dr Lechner

Thank you for submitting your manuscript to PLOS ONE. After careful consideration, we feel that it has merit but does not fully meet PLOS ONE’s publication criteria as it currently stands. Therefore, we invite you to submit a revised version of the manuscript that addresses the points raised during the review process.

We would appreciate receiving your revised manuscript by 13/04/2020. To enhance the reproducibility of your results, we recommend that if applicable you deposit your laboratory protocols in protocols.io, where a protocol can be assigned its own identifier (DOI) such that it can be cited independently in the future. For instructions see: http://journals.plos.org/plosone/s/submission-guidelines#loc-laboratory-protocols

We look forward to receiving your revised manuscript.

Kind regards,

Karen-Leigh Edward

Academic Editor

PLOS ONE

Journal Requirements:

Reviewers' comments:

Reviewer's Responses to Questions

**Comments to the Author**

1. Is the manuscript technically sound, and do the data support the conclusions?

Reviewer #1: No

Reviewer #2: Partly

Reviewer #3: Yes

2. Has the statistical analysis been performed appropriately and rigorously? 

Reviewer #1: No

Reviewer #2: Yes

Reviewer #3: Yes

3. Have the authors made all data underlying the findings in their manuscript fully available?

Reviewer #1: Yes

Reviewer #2: No

Reviewer #3: Yes

4. Is the manuscript presented in an intelligible fashion and written in standard English?

Reviewer #1: Yes

Reviewer #2: Yes

Reviewer #3: Yes

5. Review Comments to the Author

Reviewer #1: First of all, thanks for giving me the possibility to review this interesting article.

The authors intend to evaluate whether a simulation based Difficult Conversations Workshop for neonatal nurse practitioners leads to improved skills in conducting difficult conversations.

The topic is relevant and current for nursing practice in the context of NICU. The article is well written. However, in my opinion, there are some considerations that authors should consider so that the article can be published.

On the one hand, the sample size is too small for results support the conclusions and to be able to make inference. Should Authors consider it as a pilot study?

On the other hand, the study has another important limitation related to the design. Authors state that it is a randomized and controlled study with a control and intervention groups . However, test scenarios are only measured once in each group, and at different times. This does not allow to know and compare the baseline situation of both groups, nor if there are intragroup differences. I feel that authors did not evaluate the effectiveness of the Workshop through the evaluators, but the structure of the intervention ¿test scenario before or after ?. Regarding the evaluation tool, authors could explain how it was created or developed (Expert consensus?)

Another aspect to consider is that I think the introduction is too concise. The authors can go a little deeper into other intervention studies with the same objective, as well as justify why they conduct the workshop in their context. In addition, I believe that they should present a little more about the context (USA) where the study is being carried out since nursing is not the same throughout the world and may not be well understood.

Finally, I believe that the writing of the Results can improve. The authors section Results in many subsections and star each with a kind of conclusion. Some subsections only continue one line. I think it would be better to regroup the results in less subsections to facilitate reading.

I hope these comments are helpful. I encourage you to continue your work.

Best regards.

Reviewer #2: The difficult workshop time period for the lecture and simulation is not well explained. The communication skill development needs adequate time but it is not clear. Please revise

The workshop includes lecture and simulation but the discussion was focused on only simulation. Please review

The conclusion is not clear and it doesn't go with the objective of the study.Please ammend

Reviewer #3: Generally, the manuscript is well written with a good protocol. I will strongly recommend authors to look thoroughly comments and incorporate them.

Describe abbreviation fully when you are using them for the first time.

**Overall comments**

Generally, the manuscrpit is well written with a good protocol. I will strongly recommend authors to look at the following comments and incorporate them.

Describe abbreviation fully when you are using them for the first time.

**Introduction **

The introduction part doesn’t show gap. You need to show the gap between existing situation and what the situation should be.

Also, show the severity of the problem on the parents and health care services.

**Materials and Methods**

Clearly state your measurements.

Describe also reliablity and validity of your measurements.

**Conclusion**

Describe the limitations of your study under this part.

6. PLOS authors have the option to publish the peer review history of their article (what does this mean?). If published, this will include your full peer review and any attached files.

Reviewer #1: No

Reviewer #2: Yes: Mikiyas Amare Getu

Reviewer #3: No

---

## [Author Response · Author response to Decision Letter 0]

12 Feb 2020

Dr. Joerg Heber

Editor in Chief

PLOS One

 January 30, 2020

Dear Dr. Heber,

We are pleased that you have allowed us to resubmit a revised version of our manuscript “A simulation based difficult conversations intervention for NICU nurse practitioners: A randomized controlled trial” to PLOS One for consideration for publication.

We thank the reviewers for their encouraging comments and their thorough and helpful critique. We have carefully considered their comments and have made changes based on them that we believe improve the overall manuscript. Most noteworthy is the expansion of the Introduction. Please find below a point-by-point response to the Reviewer’s critique.

Reviewer 1

Critique 1: “…the sample size is too small for results support the conclusions and to be able to make inference. Should Authors consider it as a pilot study?”

The description of this study as a pilot study would not be appropriate given the definition of a pilot study as a study that is performed to test the methods and other practical aspects of a study (van Teijlingen and Hundley 2001). While the sample size is smaller than we would have liked, it is large enough that appropriate statistical analysis could be performed. Additionally, studies such as this are very difficult to do given the significant time commitment that is demanded of each subject to take part in the study, especially as the subjects are professionals with a clinical workload. Thus, there is an understandable paucity of this type of literature, making this study and our findings especially valuable to the neonatal research community.

Critique 2: “Authors state that it is a randomized and controlled study with a control and intervention groups. However, test scenarios are only measured once in each group, and at different times. This does not allow to know and compare the baseline situation of both groups, nor if there are intragroup differences. I feel that authors did not evaluate the effectiveness of the Workshop through the evaluators, but the structure of the intervention ¿test scenario before or after?”

The Reviewer’s observation that the baseline situation in both groups and intragroup differences were not measured is correct. In an ideal experimental structure, one would have a Test Scenario for both the control and intervention group, in order to compare the baseline. Then, the control group would immediately participate in a second Test Scenario, and the intervention group would take part in the Difficult Conversations Workshop and then participate in the second Test Scenario. However, in a real world clinical setting, this is not a practicable structure for two reasons. First, the study volunteers were busy professionals who were donating their time to this research study. It would have been ethically inappropriate to not allow the control group to also benefit from the teaching opportunity provided by the workshop. Second, this structure would have taken up even more of the donated clinical time of the intervention group. The study team felt that the minimal additional benefit of having a baseline comparison as well as the lack of a research advantage to not allowing the control group to participate in the workshop after they had completed their role as controls didn’t warrant using this structure for the study. Evaluating for intragroup (individual) differences was not a stated objective of the study and would not have been helpful in answering the study question of whether the Difficult Conversations Workshop improves communication skills. As the Reviewer correctly states, we evaluated the difference between “test scenario before or after”, which means, in other words, “test scenario without completed Workshop and test scenario with completed Workshop”, thus evaluating the impact that having taken part in the Workshop had on test scenario performance, which was the stated objective of the study. We have explained this experimental structure in the Materials and Methods Study structure section (page 5, lines 101-113) as well as in Figure 1.

Critique 3: “Regarding the evaluation tool, authors could explain how it was created or developed (Expert consensus?)”

Thank you for pointing out that we had not adequately described the methods for the development of the tools used. We explained the steps taken in the manuscript (page 7, line 144-147).

Critique 4: “Another aspect to consider is that I think the introduction is too concise. The authors can go a little deeper into other intervention studies with the same objective, as well as justify why they conduct the workshop in their context. In addition, I believe that they should present a little more about the context (USA) where the study is being carried out since nursing is not the same throughout the world and may not be well understood.”

We thank the Reviewer for drawing our attention to our oversight regarding the context of nurse practitioners in the US. We have added this context to the Introduction as well as expanding on similar studies in the literature both in medicine in general and in neonatology in particular (page 4, lines 54-58 and 59-64).

Critique 5: “Finally, I believe that the writing of the Results can improve. The authors section Results in many subsections and star each with a kind of conclusion. Some subsections only continue one line. I think it would be better to regroup the results in less subsections to facilitate reading.”

We have reorganized the Results section into fewer subsections to facilitate the flow of the section.

Reviewer 2

Critique 1: “The difficult workshop time period for the lecture and simulation is not well explained. The communication skill development needs adequate time but it is not clear.”

Thank you for pointing out this oversight. We have explained the timeline of the lecture and simulation aspects of the workshop as well as the entire workshop timeline in the manuscript (page 6, lines 118-126).

Critique 2: “The workshop includes lecture and simulation but the discussion was focused on only simulation. Please review.”

We thank the Reviewer for pointing out that we did not mention the lecture portion of the workshop in our discussion. We have revised the Discussion to clarify that the workshop consisted of both components (page 5, line 77; pages 12-17, lines 231-325).

Critique 3: “The conclusion is not clear and it doesn't go with the objective of the study. Please amend.”

We have revised the conclusions both in the Abstract and in the Discussion to align with the stated hypothesis (page 3, lines 46-48; page 17, lines 325-328).

Reviewer 3

Critique 1: “Describe abbreviation fully when you are using them for the first time.”

We have clarified abbreviations when using them for the first time in the manuscript.

Critique 2: “The introduction part doesn’t show gap. You need to show the gap between existing situation and what the situation should be. Also, show the severity of the problem on the parents and health care services.”

We have added literature on what is known and what is not known about the subject and where the gap is, as well as discussing the importance of communication skills in the NICU given the significant mental health. Issues parents face given the difficulty of having an infant in the NICU (page 4, lines 59-64 and 72-74).

Critique 3: “Clearly state your measurements. Describe also reliability and validity of your measurements.”

We expanded on the explanation of the measurements under (page 7, line 141-147). The reliability was tested via interrater reliability testing and found to be appropriate (page 9, lines 165-167). Validity was determined as described under (page 7, lines 144-147). In brief, the instruments were internally developed specifically for this purpose and items chosen resulted from a literature search and expert consensus on their face validity.

Critique 4: “Conclusion: Describe the limitations of your study under this part.”

The limitations of the study are described between page 15, line 292 and page 16, line 314.

We would like to thank the Reviewers again. We hope you will find the revised manuscript acceptable for publication in PLOS One.

Sincerely,

Beatrice Lechner MD

---

## [Editor Report · Decision Letter 1]

19 Feb 2020

A simulation based difficult conversations intervention for neonatal intensive care unit nurse practitioners: A randomized controlled trial

PONE-D-19-33859R1

Dear Dr. Lechner,

We are pleased to inform you that your manuscript has been judged scientifically suitable for publication and will be formally accepted for publication once it complies with all outstanding technical requirements.

With kind regards,

Karen-Leigh Edward

Academic Editor

PLOS ONE

---

## [Editor Report · Acceptance letter]

25 Feb 2020

PONE-D-19-33859R1 

A simulation based difficult conversations intervention for neonatal intensive care unit nurse practitioners: A randomized controlled trial 

Dear Dr. Lechner:

I am pleased to inform you that your manuscript has been deemed suitable for publication in PLOS ONE. Congratulations! Your manuscript is now with our production department. 

With kind regards,

on behalf of

Professor Karen-Leigh Edward 

Academic Editor

PLOS ONE